# Laboratory Results of a Real-Time SHM Integrated System on a P180 Full-Scale Wing-Box Section

**DOI:** 10.3390/s23156735

**Published:** 2023-07-27

**Authors:** Monica Ciminello, Bogdan Sikorski, Bernardino Galasso, Lorenzo Pellone, Umberto Mercurio, Gianvito Apuleo, Daniele Cirio, Laura Bosco, Aniello Cozzolino, Iddo Kressel, Shay Shoham, Moshe Tur, Antonio Concilio

**Affiliations:** 1Adaptive Structures Division, The Italian Aerospace Research Centre (CIRA), 81043 Capua, Italy; m.ciminello@cira.it (M.C.); b.sikorski@cira.it (B.S.); b.galasso@cira.it (B.G.); l.pellone@cira.it (L.P.); u.mercurio@cira.it (U.M.); 2Research Division, Piaggio Aerospace Industries, 81043 Capua, Italy; gapuleo@piaggioaerospace.it (G.A.); dcirio@piaggioaerospace.it (D.C.); lbosco@piaggioaerospace.it (L.B.); acozzolino@piaggioaerospace.it (A.C.); 3Advanced Structural Technologies, Engineering Center, Israel Aerospace Industries (IAI), Ben Gurion International Airport, Tel Aviv 70100, Israel; ikressel@iai.co.il (I.K.); sshoham@iai.co.il (S.S.); 4School of Electrical Engineering, Tel Aviv University (TAU), Tel Aviv 69978, Israel; tur@tauex.tau.ac.il

**Keywords:** structural health monitoring, real-time processing, composite structures, sensors, damage characterization, smart devices

## Abstract

The final objective of the study herein reported is the preliminary evaluation of the capability of an original, real-time SHM system applied to a full-scale wing-box section as a significant aircraft component, during an experimental campaign carried out at the Piaggio Lab in Villanova D’Albenga, Italy. In previous works, the authors have shown that such a system could be applied to composite beams, to reveal damage along the bonding line between a longitudinal stiffening element and the cap. Utilizing a suitable scaling process, such work has then been exported to more complex components, in order to confirm the outcomes that were already achieved, and, possibly, expanding the considerations that should drive the project towards an actual implementation of the proposed architecture. Relevant topics dealt with in this publication concern the application of the structural health monitoring system to different temperature ranges, by taking advantage of a climatic room operating at the Piaggio sites, and the contemporary use of several algorithms for real-time elaborations. Besides the real-time characteristics already introduced and discussed previously, such further steps are essential for applying the proposed architecture on board an aircraft, and to increase reliability aspects by accessing the possibility of comparing different information derived from different sources. The activities herein reported have been carried out within the Italian segment of the RESUME project, a joint co-operation between the Ministry of Defense of Israel and the Ministry of Defense of Italy.

## 1. Introduction

The increased use of UAV, a clear tendency that has been consolidated in the latest years for both military and civil applications, has led to the introduction of novel regulations particularly aimed at ensuring safe operations over populated areas. Among the most popular airworthiness requirements directed towards this objective, STANAG 4671 [1] may be cited as moving away from usual commercial manned aircraft regulations. The normative, specifically produced for military UAV systems with a maximum take-off weight of up to 20 tons, is intended to operate regularly in non-segregated airspace. Consistent with this view, the use of structural health monitoring, or SHM systems, is an attractive option for increasing the awareness of vehicle integrity, above all if carried out continuously for the time period. Such systems may have important impacts on maintenance actions, moving the attention to “on-demand” from scheduled actions, and, therefore, impacting the economics even more than the safety aspects. One of the expectations is that SHM can qualify as a repeatable and reliable non-destructive inspection technique, similar to those referred to in many standards and regulations. This may be particularly important for bonded structures. While the associated technology may actually lead to relevant consequences in reducing weight and manufacturing times, current protocols are, reasonably, tougher for bonded joint failures which may lead to catastrophic loss of the aircraft. In that case, it is, in fact, required that the structure is capable of bearing the design load, to whatever extent, under the maximum debonding occurrences. Alternatively, it may be decided that any associated component of an endangered mechanical link undergo an experimental verification test. The outcome is, however, expensive in terms of associated weight, and even manufacturing times: the reference to damaged structures makes them much more robust than they were, should they be designed only with reference to the external loads, by simply thickening them or adding redundant fasteners. It should also be added that drilling composites is not that easy and may, in turn, cause the onset of initial cracks. Even considering advanced NDI techniques, some particular discontinuities are difficult to be detected, such as the lack of adhesive between two interfaces, perfectly matching each other, and leaving no room for the presence of bonding agents. Furthermore, this would be hard to implement for barely accessible parts. In this case, it is relevant that the same regulation bodies, as in [2], point out the option most open to future developments and progress of NDI technologies, indicating SHM as a promising strategy.

The concept of structural health monitoring has been discussed for many years. The basic point is directly associated with the nature of structures. Basically, they are infinite-DOF systems, therefore, ideally, infinite sensors are necessary to properly observe their conditions. In turn, this means a huge quantity of cabling and, associated with the exceptional amount of expected data, very large computation memory. In this paper, the aspect of the sensor network is considered. Before proceeding, it should be remarked that a considerable complexity reduction may be achieved if the concept of hot spot is pursued [3,4,5] vs. a completely diffused network [6,7,8,9]. In many cases, the former is a viable solution, since it is known that some parts of the aircraft, or indeed of any structure, are more exposed than others to the risks of rupture. A classic example is the access door to the cabin, which experiences hits of many types while on the ground, regardless of aerodynamic actions [10]. Many systems have been proposed in the literature, specifically focusing on that region. In spite of this reduction, the problem remains, since even a section of just a few square meters, or with a perimeter of a few meters, may require hundreds of measurement points [11,12,13]. In synthesis, it can be stated that almost always the selected sensing architecture depends essentially on the problem that is to be addressed. So, the considerations that are herein described attempt a general validity by referring to systems for which extended monitoring is necessary.

Among the available SHM technologies, valuable for the above-mentioned applications, the use of distributed fiber optics sensing [14], or of multiplexed Fiber Bragg Gratings, or FBG [15], appears quite attractive. At the current status of the technology, if airborne applications are explored the secondary ones are preferred since these may count on interrogators that can be embarked without being affected by the harsh environment of the flight, or, in turn, giving effect to the on-board electronics. By allowing, generally, 16 sensors and per single fiber (also referred to as a channel), significant savings are achieved in terms of weight reduction, cabling facilitations, installation times, and system complexity. In the aerospace field, fiber optic sensors (FOS) lead to many other advantages that outperform their conventional counterparts: they are quite flexible, tolerant to environmental conditions, and transparent to electromagnetic interferences. Finally, their transversal section allows for easy embedding within large composite-material-based structural components, such as many current wings [16]. Examples of in-flight applications can be easily found in the literature. For instance, embedded FBG sensors were successfully applied on the Nishant, an Indian UAV [16]. The embedding process took place during the manufacturing process; flight data were retrieved directly on board during operation and then elaborated to detect deviations from the expected behavior. In detail, the gained information allowed the estimation of the external forces and the vibration field through analyses implemented via principal component analysis (PCA) and artificial neural network (ANN) approaches. Other studies referred to the use of FBG arrays for monitoring the structural strain in order to promptly identify variations in the wing architecture by comparing the attained data with a reference numerical model [16,17,18]. A relevant application reported the use of a 54 FBG network for a medium altitude long-endurance unmanned aerial vehicle (MALE UAV). The sensors were embedded along the wing and the tail booms to detect static and dynamic structural responses and to derive the incumbent external forces. Acquired flight data of over more than 1000 h were elaborated in both frequency and time domains to extrapolate the structural behavior and identify variations along its history. A wide review of recent applications of sensing technologies for SHM systems employing embedded sensors shows an interesting analysis of FBG capabilities with respect to other devices mounted on aircraft and other structural systems [19]. A more general scenario is reported in [20], where a number of significant examples are cited. In detail, the authors refer to the distinction among passive or active methods, depending on the characteristics of the methodology to rely on the response of the structural system as it is [21] or introducing some external energy to drive specific actions to induce some specific effect able to be measured by a dedicated network [22]. This field contains the guided waves techniques, basically aimed at using generated waves to detect irregularities in the transmission and reflection paths [23]. Concerning active systems, it should be mentioned that some attention is devoted to using existing device networks to investigate other aircraft systems’ health, such as electromechanical actuators [24].

## 2. Aims and Motivations of the Research

Any damaged condition is a rare occurrence for mechanical systems, as it is very unlikely to be observed. Thus, it represents an extreme deviation from the median of its probability distribution. It is, therefore, necessary to apply proper statistical solutions. For instance, a recent methodology for the SHM of aircraft wings has recently been proposed to perform mode shape-based damage detection [25]. On the other hand, the algorithm presented herein is based on a non-model-based damage identification approach, which implements strain shape-based damage detection. Damage features are extracted from the current strain profile of the FBG array. The basic core of the mathematical process is the assumption that structural damage is always correlated to significant strain gradient variations [26]. It moves away from edge detection techniques that are usually utilized in digital image processing for identifying the boundary of different subjects [27]. After having preliminarily proven the feasibility of such a methodology on predominantly 1D structural elements, albeit with a complex architecture [28,29], the activity reported deals with an application on a full-scale wing-box section. Like the former study, the target of investigation is the real-time detection of skin–spar debonding.

The software system implemented allows managing data retrieved through a four-channel interrogator, in principle, permitting simultaneous computation by an arbitrary number of algorithms, addressing the detection of possible structural irregularities. In detail, SHM codes developed by the Italian Aerospace Research Centre (CIRA), the Israel Aerospace Industries (IAI), and Tel Aviv University (TAU) are contemporarily executed, operating on the same record packages in real time. The output is handled to expose some visual analytics and alerts to the external operator, either acoustic or visual, that can invoke further actions to protect the aircraft. The entire computational cycle is performed in approximately one second, giving the measure of the actual real-time meaning in this application. The test article is a full-scale wing-box section, of 1.2 m span, and characterized by three main longitudinal spars. The bottom skin panel is bonded to the spar caps; the structural health monitoring methodology herein presented targets to evaluate the possible presence of irregularities, i.e., detachments, caused by any factor (such as, for instance, but not limited to, external forces, poor manufacturing, and so on), detected along those bonding lines by a dedicated deformations data-processing analysis. The reference structure is placed in a thermal chamber allowing for room temperature (RT) and high-temperature (HT) measurements, and is excited by quasi-static forces acting in the vertical and span-wise directions, produced by electromechanical actuators each capable of up to 70 kN, activated one at a time. In this way, both bending and compression states are produced, characterized by a maximum strain well over one thousand, and hundreds, of microstrains, respectively, corresponding to different operational levels. The SHM system provides recursive and continuous indications of the structural state, updating its output approximately each second, eventually accompanied by an acoustical signal if a discontinuity is detected. In synthesis, the main aims of this activity are:confirmation of the real-time functionality of the proposed SHM system already verified in [28];confirmation of the capability of the developed SHM algorithm on a more complex structure, by referring to embedded and bonded optical fibers equipped with FBG arrays;confirmation of the capability of the SHM system to handle several algorithms, at the same time, being simultaneously fed with the same data;establishment of preliminary criteria for discerning the required sensor density for the specific selected sensor arrays, in terms of number of sensible devices per unit of length; this measure is expected to be strictly dependent on the irregularity size.

## 3. SHM SW: Logic and Implementation

The local high-edge onset (LHEO) algorithm considers structural damage as an edge discontinuity along the strain energy signature. The methodology logic diagram and other details can be found in [28].

The main idea is that the onset of edge signals can be tracked in both the space and time domains, correlating the measured rate of similarity of current strain values closest to one by using an inner product as a function of the sensor gap (minimum lag of two consecutive sensors at a certain instant), and, also, as a function of the time step (minimum lag of two consecutive time acquisitions at a sensor position). By considering strain measurements (ε) as input signals, the first mathematical operation to be applied is the first derivative along the rows and along the columns of the input file (namely, horizontal and vertical derivative filters). Since, by definition, an edge is a place of rapid change in the intensity function of the signal, the edges correspond to the picks of derivative, but the difference filters also respond strongly to noise. So, to face this issue and improve the signal-to-noise ratio, the gradient vectors of the provided horizontal and vertical derivative filters are correlated at each sensor position. If cross-correlation magnitude at a pixel exceeds a threshold, a possible edge point is reposted. In detail, the cross-correlation of the strain gradient vector (dε) can be written [28] by using Equation (1) as follows:(1)$$R_{i,i+1}(t)=\frac{1}{N}\sum_{i=1}^{N-1}{d\varepsilon }_{i}\left(t\right){d\varepsilon }_{i+1}\left(t+∆\tau \right)$$
for i spanning the total number of sensors, N. If the strain gradient at the current sensor is not affected by any variation with respect to time, or, similarly, when the strain gradient signal at consecutives sensors coincide, the value of Equation (1) is maximized, and corresponds to the auto-correlation R_max_.

The relative change of normalized cross-correlation function with respect to the reference auto-correlation vector (2) is defined as a damage index as follows:(2)$${CDI}_{i}=Norm[\left(R_{i,i+1}\right)-[R_{max}]$$

Hence, in the absence of a jump/edge, the cumulative damage index (2) will be small. On the contrary, if an edge is present, then the two function values in (2) will be quite different from zero [28].

Since this methodology is based on relative comparisons and correlations, it is able to provide local information despite a global one. Of course, the output is affected by how the energy distribution is spread over the structure and is, naturally, affected by the presence of strain concentration areas as load application points or constraint points. This is the reason why, despite the fact that Equation (2) could be still considered the estimation of the highest true positive occurrences, it is useful to introduce a cut-off value acting as a lower limit level. The operation of setting the cut-off is a sort of tuning which can be conducted considering whatever reference condition applies. In fact, it is useful to introduce a cut-off value acting as a lower limit level. Readouts below this lower limit are discarded. In this case, the TL cut-off value is arbitrarily chosen to be the mean value from the Equation (2).

It is also important to remark that in the considered experimental specimen, damage areas were initially introduced during its manufacturing. Nevertheless, if faults occur during the operation of the structure, the SHM system is expected to still be effective since the proposed approach works on relative changes in the strain signal. So, the onset of damage, whenever it happens, is characterized by sufficient length or energy, and is always linked to a detectable discontinuity, or to a strain edge.

## 4. Test Preparation and Test Article Description, Including the Loading System

The purpose of this paragraph is to provide information about loads, test preparation, and the procedure for a static test involving a composite small wing-box demonstrator manufactured and integrated with some FBG optical fibers. More FBG fibers are deployed externally on the bottom panel (i.e., the one that is bonded to the main structure, consisting of the top panel and three longitudinal spars) at locations that are defined at relevant sections. In total, seven flaws are artificially located between the three spars and the bottom panel. These positions will also be recalled.

The main goals of the test are: to verify SHM algorithm capabilities to detect damage when temperature gradient occurs;to verify the real-time data management capability;to perform SHM tests on a representative ground test article.

In Figure 1, a view of the naked box is reported, before the imposition of the artificial de-bonding between the bottom panel and the spar caps, the deployment of the fibers, and the gluing of the two parts and the bottom panel.

The small composite wing-box demonstrator is figuratively extracted from a complete wing-box, as detailed in Figure 2. In particular, the test article is representative of a wing segment between the stations (approximately), BL = 4100 and BL = 5200. The overall dimensions of the test article follow, as reported in Table 1, and shown in Figure 2, for a better understanding:

For the test execution, the test article (TA) is installed on the load rig in a laboratory environment, see Figure 3, with the lower skin facing up. This choice renders the installation and handling of the sensors easier. The TA is subjected to static tests in two different load conditions, namely, compression and bending, at different temperatures (room temperature and around 70 °C). To that aim, it is installed within a walk-in climatic chamber, specifically exploring the heating effect over the data processing. In this first investigation outside the environment temperature, it was chosen to select the most affordable high-temperature range (in terms of costs and complexity of the necessary facility), with the aim to move afterwards to the more challenging interval of the cold conditions.

Loads are applied by means of two different hydraulic actuators, each for the two configurations prescribed, see Figure 4 (schematic) and Figure 3 (lab picture). The hydraulic actuators are controlled by means of a cyber servo controller, regulating the hydraulic oil pressure in the cylinders, based on the actual force acquired through a load cell. The main characteristics of the two actuators are reported below, see Table 2. The root of the small composite wing-box is fixed on a rig beam by means of a set of machined plates and bolts, properly sized, while the hydraulic jacks are constrained to a rig beam positioned above the rig for the bending test and to another lateral rig beam for the compression test. The test article is, therefore, in two configurations:bending: the load FB, applied along the opposite direction of the Z axis (aircraft axis);compression: the load FC, applied along the opposite direction of the Y axis (aircraft axis).

The reference limit load for both conditions is FC = FB = 20,000 N. At the end of the limit load cycles, it is decided to go up to the jack full scale in bending condition at room temperature, with steps of a 10% LL (approaching ultimate load (UL)). For the bending test, the hydraulic jack pulled while the jack pushed for the compression test.

To show that the temperature distribution is constant over the entire surface of the test article, pictures were taken with a Testo thermal imager and reported below, see Figure 5.

## 5. Sensors Layout and Damage Locations

The test article is supplied for test laboratory with four embedded optical fibers (OF), and seven de-bonds, applied as reported in Figure 6. Flaws and inner OF are, therefore, placed between the wing-box main body and the bottom panel. Inner OF and flaws positions are listed in Table 3. The spar print (i.e., the shape of the caps bonded to its internal surfaces) are shown in red dotted lines. The embedded fibers were placed in pairs for redundancy, to minimize the possibility of breakage during the manufacturing process that could have prevented strain acquisitions. In any case, all four fibers survived.

Five further optical fibers are bonded on the outer side of the bottom skin, in order to verify the monitoring capability when the sensors are located far from the bonding line. The region of interest, i.e., the region where the bonding of the optical fibers takes place, is graphically reported in Figure 7. There, a schematic of the region reports where the fibers have been deployed, with an indication of the flaws and the optical elements themselves, for better clarity.

The SHM algorithm readouts are summarized in the following tables. The embedded fibers A and B, and C and D are twins; this means that they have the same layout in terms of the number of FBG sensors and their relative positions along the spar. Such embedded FO are arrays of 11 (A and B) and 8 (C and D) FBGs, respectively, having a space resolution of 20 mm. The bonded fibers, on the contrary, have different layouts and each of them is used to monitor different damage areas by using different FBG space resolutions.

The bonded FO are arrays of 16 FBG but the space resolution is not the same for all fibers. For the sake of simplicity, the fibers are colored in red, black, and blue. The red and blue fibers are used to monitor a single region of interest, 80 mm long, with a 20 or 10 mm space resolution for the red and blue fibers, respectively (Figure 7). The black fiber is 1000 mm long and is used to monitor the whole length of the spar. In this case, the space resolution is 50 mm.

For the sake of clarity, two sketches of the wing-box reporting the OF layout and the damage positions are provided. The OF labelled A and B, and C and D are associated to the front and rear spar, respectively, and are deployed within and along the bonding line (see Figure 6). The second picture, Figure 7, presents the surface-bonded optical fibers. In this case, labels 01, 02, and 05 for the front spar, and 03 and 04 labels for the rear spar, are adopted. The fibers 04 and 05 (black dashed lines) cover the whole length of the spar, so they are able to provide data from damage 2, 3, and 6, respectively. The damaged 7 is never monitored.

## 6. Implementation of the HUMS Code on a Local Platform

The hardware used to run all the real-time software and the SHM algorithms is a single consumer-grade personal computer (PC)—an Alienware X17 laptop with an i7 11,800 H processor, 32 GB of RAM, and a gigabit ethernet port. The PC is running Windows 11 as the operating system (OS). The interrogator (SmartScan2) is connected to the laptop through a TP-Link Archer MR600 modem/router and all devices are configured to be on the same local address network (10.0.0.X). The TP-Link modem/router has an LTE cellular data connection providing Internet access for the PC, including remote desktop access, during the tests. A sketch of the HW setup is shown in Figure 8.

The equipment listed below is strictly necessary to run the experiment. A logic schematic of the test set-up is reported in Figure 9. In detail:optical fibers;optical fiber stand-off cables;optical interrogators;PC equipped, with;optical interrogator SW;oCIRA SHM SW;oTAU SHM SW;oreal-time decoding SW;Wi-Fi router for driving the PC and the interrogator.

All of the test runs that utilized the real-time sub-system are configured to execute three concurrent instances of SHM algorithms. This allows the verification and confirmation of the correct behavior of the whole system in its full project configuration with executables coming from different SHM codes and operating simultaneously on the same data coming from the interrogator. A multiple instance configuration permits simultaneous execution of the same SHM code on more than one sub-group of FBGs, for instance, to elaborate each fiber separately. Of course, each instance receives and operates on the same data.

The figure below, Figure 10, depicts screen captures of the computers’ desktop with ongoing real-time tests performed at Piaggio Aerospace. The image depicts the prompt DOS with the current data streaming and the corresponding strain graphs. Three contemporary SHM instances, two instances of CIRA’s SHM and one instance of TAU/IAI SHM code, are elaborating the same data in real time. The incoming data-streamed package is indicated by bracket reporting the line quantity (in this case, (1100)), and the number of sensors (in this case, (7)) for two arrays (“calc using arr1 and arr2”). For each package, the SHM outcomes provide information about the sensor number indicating a damage occurrence (“damage detected at: 2”). The reported image is aimed at illustrating the simultaneous working of the three SWs, managing two bar plots (one for each array) and one data plot of the strain vs. time for the two arrays (Ch2 and Ch3) comprised of seven FBG each (as reported in the legend).

## 7. HUMS Code Implementation

According to the test matrix defined to validate functionality and effectiveness of the methodology, the activities are planned to start with bending excitation both at room temperature and high temperature (RT and HT), and then with axial compression, again at room temperature and high temperature. An offset calibration of all the sensors is performed during the test executions before each data-logging. Since the real-time execution is storage-consuming, it is decided to realize this specific validation set just after the data-logging session is over (see previous paragraph).

***a.*** 
**
*Bending at RT/HT (embedded sensors A and B)*
**


As a first step, bending effects are recorded by the front spar fibers, A and B, at RT and HT. The figure below, Figure 11 and Figure 12, report the data-processing for a single fiber at RT (right) and HT (left), respectively, for quick comparison of the temperature variation effects over the strain. Specifically, Figure 11 reports the strain magnitude along the y-axis during the incremental loading up to LL, while the x-axis represents the index of the FBG sensors from 1 to 10. Figure 12 represents the cumulative damage index (CDI) for each sensor (x-axis) estimated from Equation (2).

The threshold level (TL), set at the value 2, is estimated as the mean value of the CDI distribution for the current data-streamed package. The SHM system indicates the occurrences of edge onset each time the sensor value exceeds the TL, hence, at sensors 6, 7, 8, and 9 at RT for fiber A and at sensors 5, 8, and 9 at HT for the same fiber A (red dots indicating the eligible sensors); similarly, at sensors 3, 4, 5, 6, 8, and 9 at RT and at sensors 4, 8, and 9 at HT for fiber B (black dots indicating the eligible sensors).

***b.*** 
**
*Compression at RT/HT (embedded sensors A and B)*
**


The compression effect is retrieved by the same fibers above, A and B, at RT and HT. The figure below, Figure 13, reports the strain magnitude along the y-axis during the incremental loading up to LL, while the x-axis reports the index of the FBG sensors from 1 to 10. The Figure 14 represents the cumulative damage index for each sensor (x-axis) estimated from Equation (2).

As the threshold level (TL) is fixed at 2, estimated as the mean value of the full dataset logged during the streaming, the CDI indicates the occurrences of edge onset each time the value exceeds the TL, hence at sensors 6, 7, 8, and 9 at RT and at sensor 6 at HT for fiber A; similarly, edge onset occurs at sensors 3, 6, 7, and 9 at RT and at sensors 3, 6, 7, and 9 at HT for fiber B.

***c.*** 
**
*Bending at RT/HT (embedded sensors C and D)*
**


Again, the same description is provided for the seven FBG array fibers C and D along the rear spar, at RT and HT. Figure 15 and Figure 16 report data processing for RT (left) and for HT (right).

***d.*** 
**
*Compression at RT/HT (embedded sensors C and D)*
**


The compression effects are then investigated, as retrieved by the fibers C and D at RT and HT. The figures below, Figure 17 and Figure 18, report the data processing for RT (left) and for HT (right).

***a.*** 
**
*Bending at RT/HT (bonded sensors 01, 02, and 05)*
**


Then, the bending effects are detected by fibers 01, 02, and 05, deployed on the surface skin in the region of the front spar, see Figure 7, at RT and at HT. The figures below, Figure 19 and Figure 20, report the data processing for RT, on the left, and for HT, on the right. Figure 19 represents the strain logged during the incremental loading up to LL, along a generic fiber. Figure 20 represents the CDI derived from the four features. Fibers 01, 02, and 05 were each equipped with 15 FBGs sensing points.

Having fixed the TL at a value of 2, as before, the CDI indicates the occurrences of an edge onset at sensors 2, 3, 4, 5, and 6 at RT and at HT for fiber 01. Such an occurrence verifies at sensors 5 and 8–12 at RT and the same plus 13 HT for fiber 02. For fiber 05, the sensors are 3, 6, 7, 9, and 12 at RT and 7, 10, 12, and 13 at HT.

***b.*** 
**
*Bending at RT/HT (bonded sensors 03 and 04)*
**


In this case, the bending effects are retrieved as represented below where Figure 21 and Figure 22 report the data processing for RT along the left column, and for HT along the right column, for quick comparison. Fibers 03 and 04 were carried on 15 FBGs sensing points.

## 8. SHM Results

The tables report the FBG indexing (geometrical order) from 1 to 10. The black cells of the table correspond to the current position of the damage. The SHM readouts are indicated by applying an X flag to the corresponding FBG box.

As the tables show, it is clear that the edge detection principle at the base of the methodology is able to detect the damage onsets (Figure 23 and Figure 24). It is also important to remark that numerical studies have demonstrated that the effect of the damage over the strain field extends larger than the current damage size [29]; therefore, it can be expected to find out readouts at the inner and outer boundaries of the damage borders.

The predictions retrieved for the bonded fibers 2 to 4 are reported in a table form, see Figure 25, Figure 26, Figure 27 and Figure 28. There, the referred fiber is recalled according to the color code previously introduced, see Figure 7. For the sake of clarity, a small picture indicating the position of the fiber accompanies each table. The surface-bonded optical fibers show how noisy the post-processing readout is with respect to the embedded sensors’ effectiveness in the true positive estimation. Possible reasons can be attributed to the bonding quality, to the spatial resolution, and to the surface irregularities of the CFRP panel.

## 9. Conclusions

The proposed system has met our expectations:real-time operations are conducted duly, using a one second sample data period;the handling of multiple SHM algorithm has been proven;the detection of damage has been verified.

In particular, the SHM algorithm, adapted for real-time, on-board installation, motivated by previously developed distributed sensing-based routines, has been checked to provide satisfactory solutions with medium to high sensor density, ranging between 1 and 2 cm. Instead, 5 cm step networks have shown some limitations, because of the following factors:the largest length of the herein investigated damage size (80 mm) is close to the step of 50 mm; therefore, just one or two sensors may be deployed in the affected region, making it difficult to identify some significant strain variation;the larger the distance between sensors, the lower the correlation level between two consecutive sensors may be clearly detected, and since the major gradient variations are expected to occur at the structural irregularity boundaries, this, in turn, can be critical.

On the other hand, the localization of the effects of the damage, widely verified in other works from the same authors, allow denser networks to catch such effects as isolated discontinuity. Furthermore, if it is not detected it means that the damage is very small and, therefore, is not seen as a threat (at least, not yet) to the health of the structure. This fact is strengthened by the consideration that the discontinuities are expected to have an effect larger than their size, therefore facilitating their detection.

An array of 16 FBG is, therefore, able to efficiently monitor a length of about 30 cm, which can be enough for the analysis of certain critical parts of the structure. Technology allows the use of even more FBG on the same fiber, and the acquisition channels may grow, opening the path to an exhaustive operation of detection with relatively low expense in terms of computation power. It should be noted that the experimental software was able to perform its duty in about 1 s, and this number can be significantly reduced by an order of magnitude using proper optimisation. Equivalently, it can be preliminarily and roughly stated that the data package size may grow by an order of magnitude, preserving the same elapsed time.

A further outcome of the reported study concerns the use of the thermal chamber. At least, in this case, it gave evidence how temperature variations barely affect the algorithm’s detection capability; in fact, since the algorithm works on strain differences among the FBG array elements, the thermal effect may be neglected since it equally affects all those sensors.

## Figures and Tables

**Figure 1 sensors-23-06735-f001:**
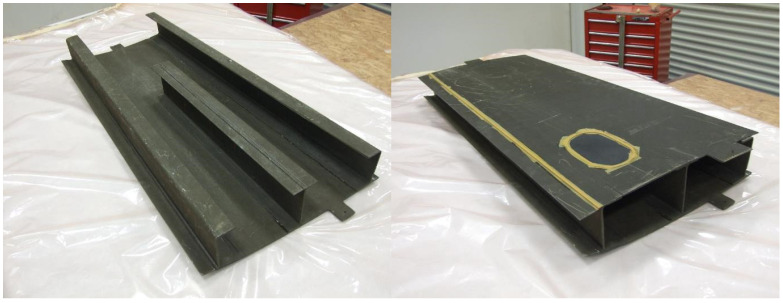
General view of a small composite wing-box demonstrator.

**Figure 2 sensors-23-06735-f002:**
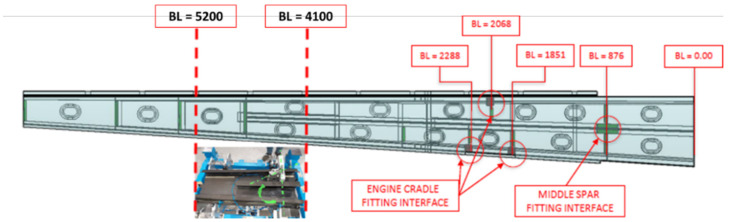
Wing-box section from which the test article has been taken.

**Figure 3 sensors-23-06735-f003:**
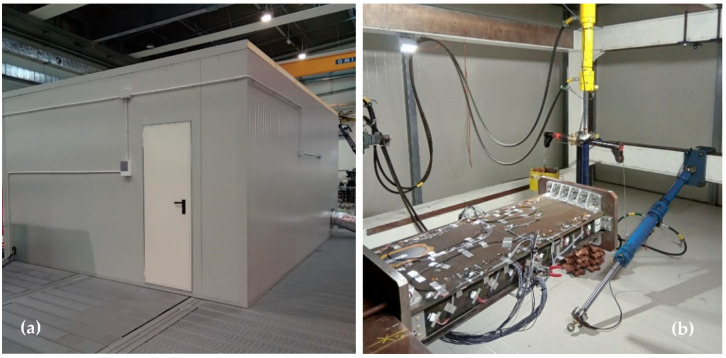
Actual test configuration: (**a**) left picture reports the climatic chamber specifically designed to host the test rig; and (**b**) right picture reports the bending actuator installed on the test article, while the compression actuator is detached.

**Figure 4 sensors-23-06735-f004:**
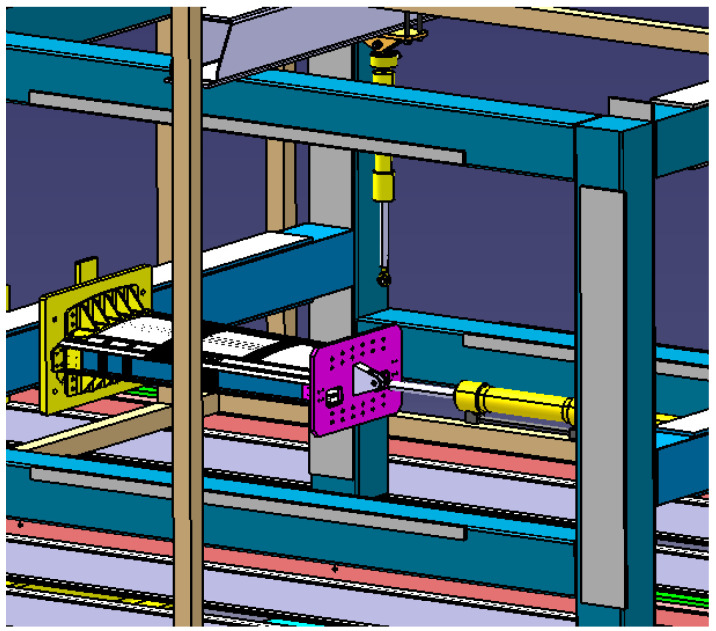
Load direction in compression and bending configuration.

**Figure 5 sensors-23-06735-f005:**
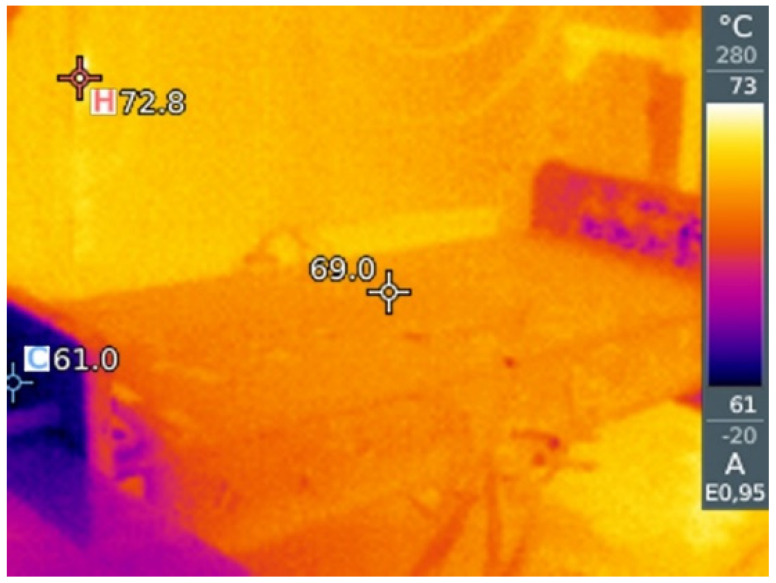
Thermal scan of the wing-box under thermal conditioning—test @ 70 °C.

**Figure 6 sensors-23-06735-f006:**
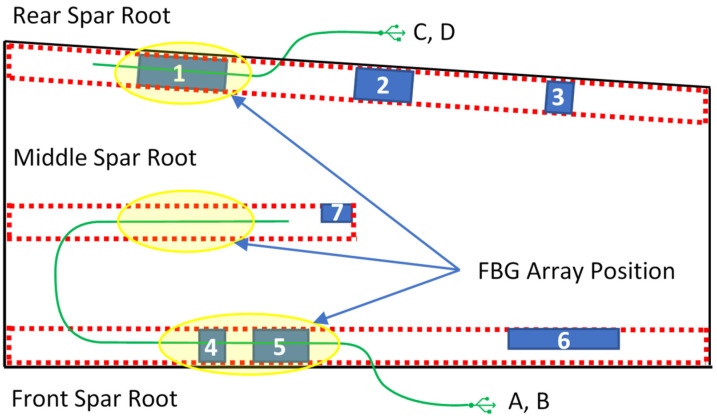
Optical fibers and damage locations—sketch. Embedded optical fibers are shown in light green and coded according the letters **A**, **B**, **C**, and **D**, while the qualitative positions of the FBG arrays are reported in the tallow oval. Damage locations are presented as blue parallelograms, and their numbering is reported in white digits.

**Figure 7 sensors-23-06735-f007:**
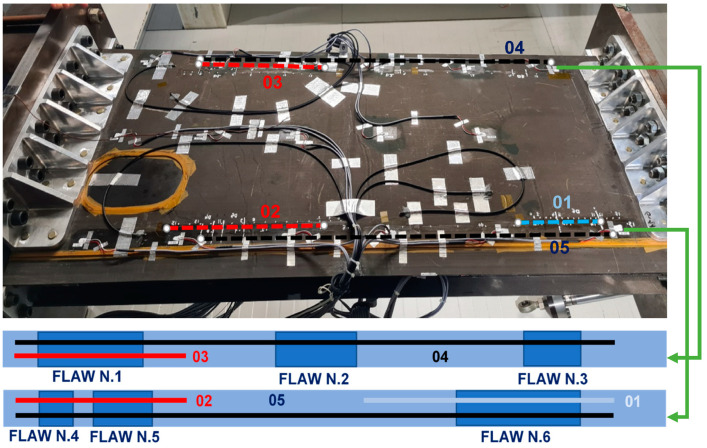
Schematic layout of the additional fibers installed on the wing-box, bottom panel, with the associated code; namely, 03 and 04 for the rear spar; and 01, 02, and 05 for the front spar. For the sake of clarity, a synthetic scheme of the flaws and fiber positions is reported, for both the spars, top and bottom. These same schematics are used to describe the results, in the last part of the paper.

**Figure 8 sensors-23-06735-f008:**
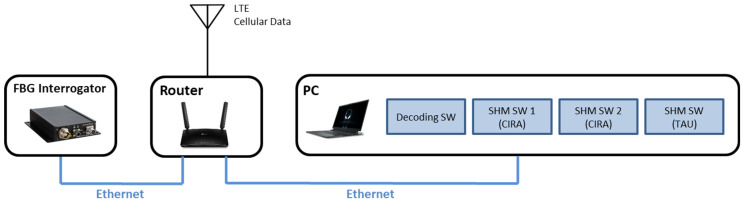
Hardware setup.

**Figure 9 sensors-23-06735-f009:**
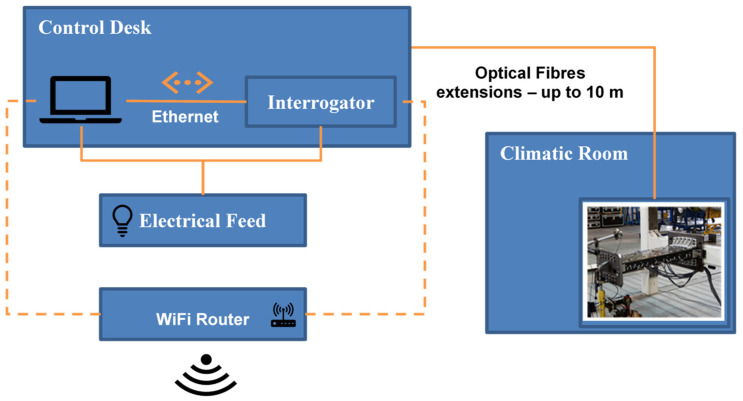
Block diagram of the experimental set-up.

**Figure 10 sensors-23-06735-f010:**
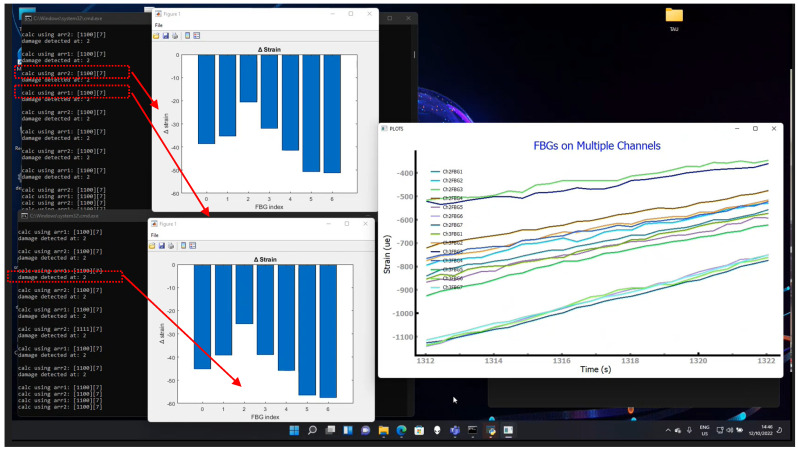
Screenshot of the SHM module running in real-time mode. Red box indicates the channel (arr1 and arr2) with the corresponding bar plot and the readout (damage detected at sensor number 2).

**Figure 11 sensors-23-06735-f011:**
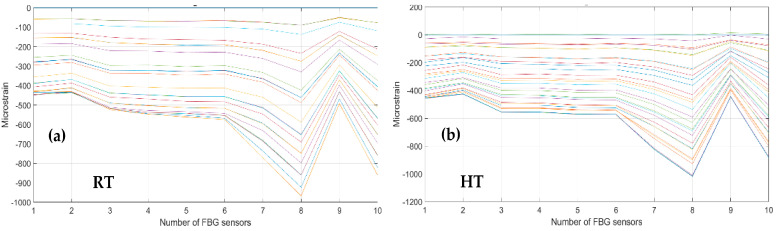
Strain distribution in the function of the increasing bending load each curve characterized by a different color. For an embedded FBG sensors array, along the front spar; (**a**) the incremental strain at room temperature; and (**b**) the incremental strain at high temperature.

**Figure 12 sensors-23-06735-f012:**
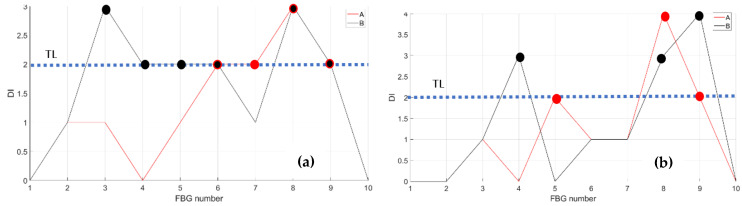
Cumulative damage index under bending: red line sensor A; black line sensor B; TL as the dashed blue line; (**a**) the cumulative damage index at room temperature; (**b**) the cumulative damage index at high temperature.

**Figure 13 sensors-23-06735-f013:**
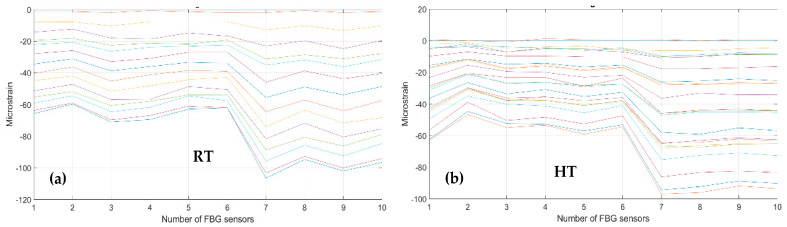
Strain distribution in the function of the increasing compression load each curve characterized by a different color. For FBG sensors embedded array, along the front spar; (**a**) the incremental strain at room temperature; (**b**) the incremental strain at high temperature.

**Figure 14 sensors-23-06735-f014:**
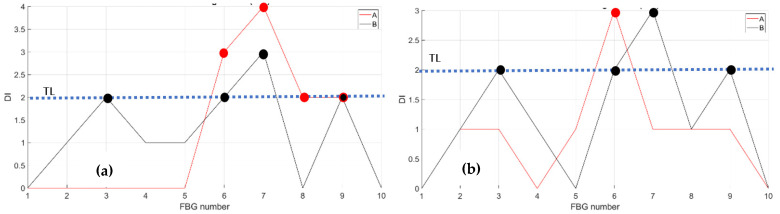
Cumulative damage index under compression: red line sensor A; black line sensor B; TL as the dashed blue line; (**a**) the cumulative damage index at room temperature; (**b**) the cumulative damage index at high temperature.

**Figure 15 sensors-23-06735-f015:**
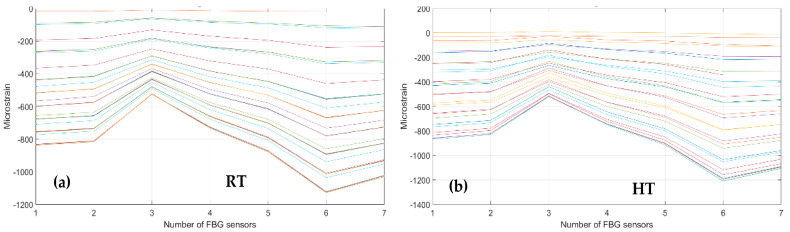
Strain distribution in the function of the increasing bending load each curve characterized by a different color. For FBG sensors embedded array, along the rear spar; (**a**) the incremental strain at room temperature; (**b**) the incremental strain at high temperature.

**Figure 16 sensors-23-06735-f016:**
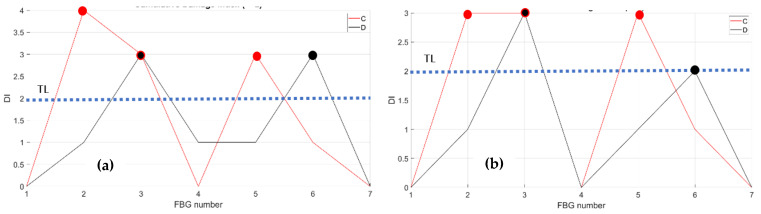
Cumulative damage index under bending: red line sensor C; black line sensor D; TL as the dashed blue line; (**a**) the cumulative damage index at room temperature; (**b**) the cumulative damage index at high temperature.

**Figure 17 sensors-23-06735-f017:**
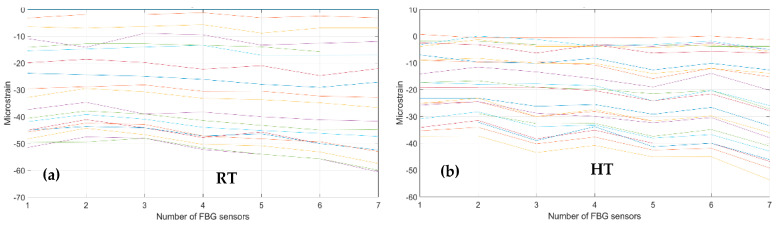
Strain distribution in the function of the increasing compression load each curve characterized by a different color. For FBG sensors embedded array, along the rear spar; (**a**) the incremental strain at room temperature; (**b**) the incremental strain at high temperature.

**Figure 18 sensors-23-06735-f018:**
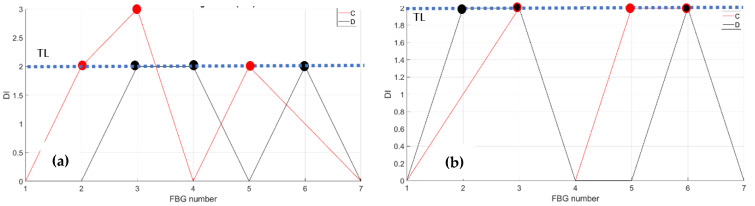
Cumulative damage index under compression: red line sensor C; black line sensor D; TL as the dashed blue line; (**a**) the cumulative damage index at room temperature; (**b**) the cumulative damage index at high temperature.

**Figure 19 sensors-23-06735-f019:**
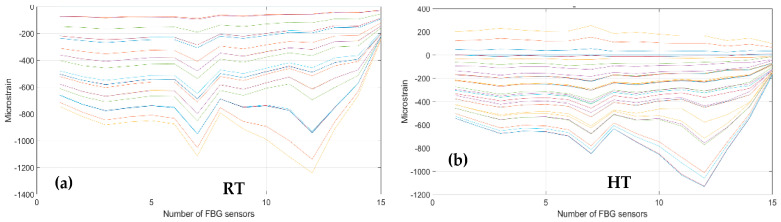
Strain distribution in the function of the increasing bending load each curve characterized by a different color. For FBG sensors bonded array, along the front spar; (**a**) the incremental strain at room temperature; (**b**) the incremental strain at high temperature.

**Figure 20 sensors-23-06735-f020:**
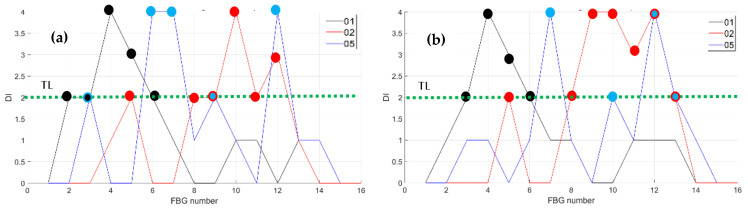
Cumulative damage index under bending: black line fiber 01; red line fiber 02; blue line fiber 05; TL as the dashed green line; (**a**) the cumulative damage index at room temperature; (**b**) the cumulative damage index at high temperature.

**Figure 21 sensors-23-06735-f021:**
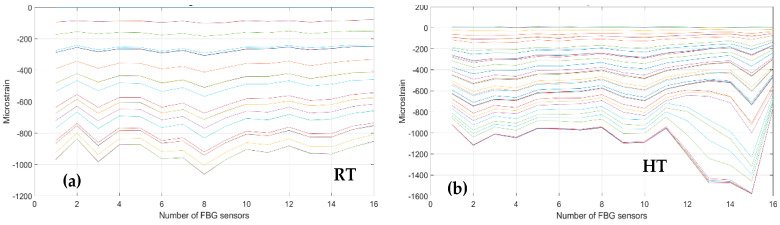
Strain distribution in the function of the increasing bending load each curve characterized by a different color. For FBG sensors bonded array, along the rear spar; (**a**) the incremental strain at room temperature; (**b**) the incremental strain at high temperature.

**Figure 22 sensors-23-06735-f022:**
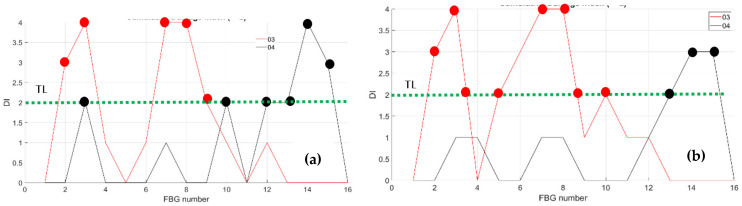
Damage index under bending: red line fiber 03; black line fiber 04, TL as the dashed green line; (**a**) the cumulative damage index at room temperature; (**b**) the cumulative damage index at high temperature.

**Figure 23 sensors-23-06735-f023:**
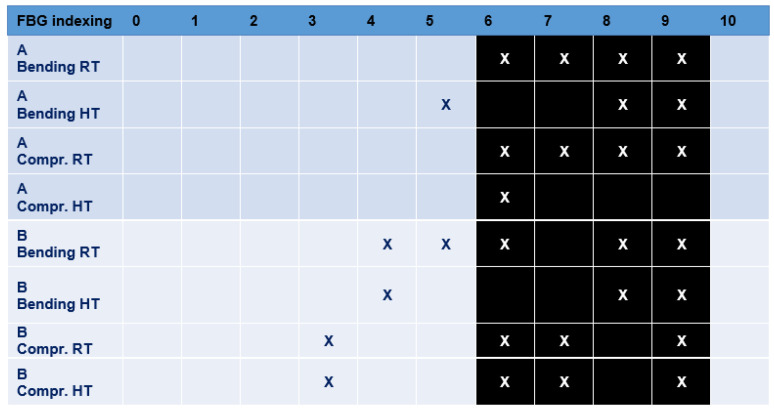
Main results for damage detection by fibers A and B. The black color refers to the position of the damage.

**Figure 24 sensors-23-06735-f024:**
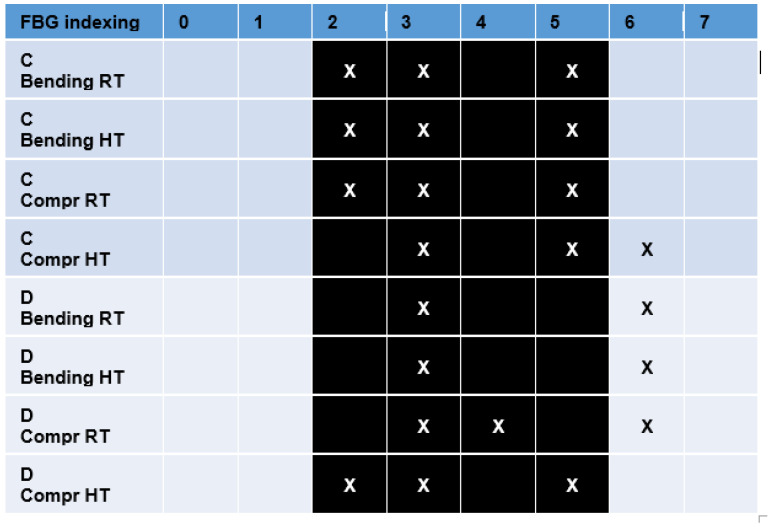
Main results for damage detection by fibers C and D. The black color refers to the position of the damage.

**Figure 25 sensors-23-06735-f025:**
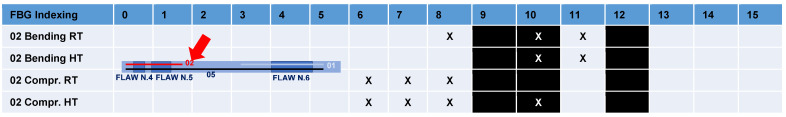
Main results for damage detection by fiber 02. The black color refers to the position of the damage. The arrow indicate the fiber under test.

**Figure 26 sensors-23-06735-f026:**
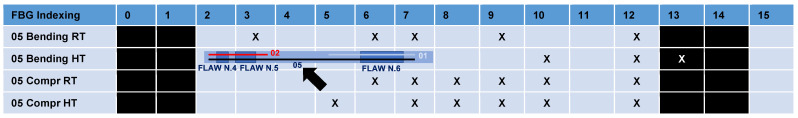
Main results for damage detection by fiber 05. The black color refers to the position of the damage. The arrow indicate the fiber under test.

**Figure 27 sensors-23-06735-f027:**
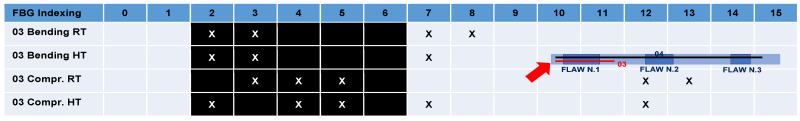
Main results for damage detection by fiber 03. The black color refers to the position of the damage. The arrow indicate the fiber under test.

**Figure 28 sensors-23-06735-f028:**
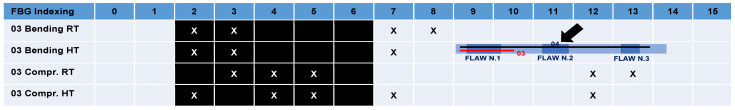
Main results for damage detection by fiber 04. The black color refers to the position of the damage. The arrow indicate the fiber under test.

**Table 1 sensors-23-06735-t001:** Main geometrical characteristics of the test article.

Length (mm)	Max Cross-Section Height (mm)	Root Chord (mm)	Tip Chord (mm)
1190	163	566	492

**Table 2 sensors-23-06735-t002:** Actuator system characteristics.

Channel ID	Actuator	Full Scale (kN)	Load Cell Full Scale (kN)	Calibration
Bending	*HD20*	*31*	*25*	*Tension*
Compression	*HDS30*	*70*	*25*	*Compression*

**Table 3 sensors-23-06735-t003:** Damage length and position along the spar.

Flaw N	Flaw Length (mm)	Flaw Width	Spar	Starting Point from “*Upper Left Corner*”
1	80	Whole cap	Front	160
2	40	Whole cap	Front	580
3	20	Whole cap	Front	890
4	20	Whole cap	Rear	240
5	40	Whole cap	Rear	280
6	80	Half a cap	Rear	880

## Data Availability

Data are not available publicly for confidentiality reasons.

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
