# Peer review of "Laboratory Results of a Real-Time SHM Integrated System on a P180 Full-Scale Wing-Box Section"

_sensors, 2023, doi:10.3390/s23156735_

Round 1

Reviewer 1 Report

The paper

‘Laboratory Results of a Real-Time SHM Integrated System on a P180 Full-Scale Wing Box Section’,

By Ciminello et al.,

Presents some relevant content in the field of Structural Health Monitoring (SHM) for aerospace applications. Specifically, the preliminary results of a real-time SHM system are reported. Importantly, the system is experimentally validated, using as a test benchmark a full-scale wing-box section and a climatic room to simulate different operating temperatures. Different SHM algorithms, developed by CIRA, IAI, and TAU have been applied on the same dataset. From this perspective, the work represents a very good example of synergy between several international research institutes and private companies.

This reviewer has no major remarks, as the concept is solid and the execution seems to have been well performed. Thus, only a few minor remarks are suggested to the Authors, to further improve the readability and the impact of their work.

MINOR REMARKS:

1.      Please add the line numbers, it helps out significantly for the peer review process.

2.      Equations should not be marked all in bold characters.

3.      Page 5: the values of length, max cross-section height, root and tip chord could be better reported in a dedicated Table rather than in a bullet point list.

4.      Figure 3: the caption should report distinctly for (a), on the left, and (b), on the right. Also, the picture on the left could be omitted or replaced, as it does not provide very useful information.

5.      Figure 4: as above, the caption should report distinctly for (a), on the left, and (b), on the right. Also, perhaps one of the two figures would be enough to describe the intended configuration.

6.      Figure 6: the caption is a bit lengthy. It would be better to report all this information in the text of the paper and use only the minimum necessary for the caption.

7.      Table 2:

8.      Figure 8 is low quality, please replace it with the same image but with a higher resolution.

9.      Figure 10 is arguably not very useful and could be omitted.

10.   Figures 11, 13, 15, 17, 19, and 21: the caption should report distinctly for (a), on the left, and (b), on the right. Also, the legends should be added.

11.   Figures 12, 14, 16, 18, 20, and 22: the caption should report distinctly for (a), on the left, and (b), on the right. Also, the caption should refer to TL as the *dashed* blue/green line, and this should be added to the legends. Finally, different line thicknesses can be used, to make the different lines visible when overlapped, and the measurement unit for DI should be added as well in the y-axis label.

12.   Tables 3 and 4: please use the journal’s template for Tables (or, alternatively, report the same graphical objects as Figures instead of Tables).

13.   Tables 5 to 8: Please do not mix tables and figures. Also, the overlapped sketches have very low quality. Please replace with versions with higher resolutions.

14.   The article is overall complete but can benefit from an expanded state-of-the-art review. For instance, a recent methodology for the SHM of aircraft wings has been recently proposed at https://doi.org/10.3390/app11041716.

15.   Page 16: generally, the use of references ([23]) should be avoided in the Conclusions.

Overall, the English is acceptable but the text can be double-checked during the proofreading stage for typos and grammar mistakes. 

Author Response

Dear reviewer,

the authors really thank you for the opportunity you offer us to improve the quality of tthe manuscript.

In the attached file, your requests are processed point by point and integrated within the paper in red colour.

best regrads

Reviewer 2 Report

The reviewed paper is devoted to a highly urgent topic of SHM-system development and validation for real-world engineering applications such as aerospace structures. In particular, strain-based monitoring relying on distributed optical sensor networks is performed for a larg-scale part of P180 aircraft wing. The paper content is definitely within the scope of Sensors journal and could be of very high interest for the scholars dealing with SHM research. 

Regarding the manuscript text, there are only some minor remarks and comments which are as follows:

1) In the introduction, it might be plausible to pay a little bit more attention to the previous works of other authors where certain results concerning the adoption of either passive or active SHM systems to real-scale constructions related to aerospace industry were presented.

2) Why high temperatures rather than the low one, e.g., below 0°C is considered? To the reviewer's opinion, the latter is more typical for real-case applications.

3) From Figure 7 it is not completely clear how these additionally installed FOS are consistent with the damage locations 2,3,6 and 7 from Figure 6.

4) In the considered experimental specimen, damages were initially introduced during its manufacturing. How would the developed SHM system perform when damages occur during the structure operation? Could the authors discuss this question somehow?

5) Page 11: "As the threshold level, TL, is set at the value 2, estimated as the mean value of the CDI  distribution..." To what extent is it reliable to evaluate TL in a way as it is described here?

Author Response

Dear reviewer,

the authors really thank you for the opportunity you offer us to improve the quality of tthe manuscript by selecting very interesting issues regarding the climatic test and treshold level estimation.

In the attached file, your requests are processed point by point and integrated within the paper in red colour.

best regrads
